# Prevalence of Depression and Anxiety in Nurses during the First Eleven Months of the COVID-19 Pandemic: A Systematic Review and Meta-Analysis

**DOI:** 10.3390/ijerph19031154

**Published:** 2022-01-20

**Authors:** Barbara Ślusarska, Grzegorz Józef Nowicki, Barbara Niedorys-Karczmarczyk, Agnieszka Chrzan-Rodak

**Affiliations:** Department of Family and Geriatric Nursing, Medical University of Lublin, Staszica 6 Str., PL-20-081 Lublin, Poland; basiaslusarska@gmail.com (B.Ś.); baskaniedorys@gmail.com (B.N.-K.); agnieszkachrzan607@op.pl (A.C.-R.)

**Keywords:** symptoms, depression symptoms, COVID-19, frontlines, nurses, meta-analysis, systematic review

## Abstract

The high risk of coronavirus (COVID-19) infection can increase the physical and psychological strain on nurses in professional practice, which can lead to mental health problems. The purpose of this systematic review and meta-analysis is to establish and estimate the combined incidence of depression and anxiety among nurses during the COVID-19 pandemic using standard measurement tools. A systematic search of the electronic databases PubMed, Web of Science, and SCOPUS was carried out to identify cross-sectional studies in the period from 3 March 2020 to 18 February 2021. Two reviewers independently and critically evaluated the studies which have been included, using the Agency for Healthcare Research and Quality checklist. We have identified twenty-three studies (*n* = 44,165) from nine countries. The combined incidence of depression among nurses was 22% (95% CI 0.15–0.30, *I*^2^ = 99.71%), and anxiety symptoms 29% (95%CI 0.18–0.40, *I*^2^ = 99.92%). No significant difference was observed in the percentage of depression and anxiety between the study subjects working on the frontlines vs. those in a mixed group (those working on the frontlines and behind the lines). This meta-analysis shows that over one-fifth of nurses in professional practice during the COVID-19 epidemic suffer from depression disorders, and almost one-third experience anxiety symptoms. This underscores the importance of providing comprehensive psychological support strategies for nurses working in pandemic conditions. Further longitudinal research is necessary to assess the severity of mental health symptoms related to the COVID-19 epidemic factor.

## 1. Introduction

The Severe Acute Respiratory Coronavirus-2 (SARS-CoV-2) pandemic has brought about many socio-economic changes in many countries; it has also triggered heavy burdens on health systems around the world, many of which had already been struggling with problems [1]. The World Health Organization (WHO) declared a global pandemic of SARS-CoV-2 on 11 March 2020 [2]. Since then, many countries have introduced limitations on social interaction and people have had to adapt to new restrictions at work and to the demands of social isolation. Nurses in their turn commenced working in the new reality of the pandemic, being on the frontlines of the fight against coronavirus, and their work plays a key role in the well-being and health of society as a whole [3,4].

Nurses have had to confront numerous problems which had, and continue to have, an effect on the quality of their work and their mental health [5,6,7]. The large number of cases increased the number of patients hospitalized with COVID-29, and often the state of health of these patients was very serious, requiring specialist care in intensive care units. Another problem is nurse staffing shortages all over the world, aggravated during the pandemic by sick leaves associated with SARS-CoV-2 infections. The already-existing nursing personnel shortage has dramatically worsened. One consequence of this has been increased workloads. There was also the problem of availability of equipment and personal protective gear. There have been situations where, due to the shortage of protective equipment, healthcare workers (HCW) faced the need to work without full epidemic protection [8,9]. Nurses worry both about the health of their loved ones, and about the risk of bringing the virus into their homes. The pandemic has caused thousands of deaths, which can be a source of existential stress [8,10].

The phenomenon of the COVID-19 pandemic and dynamic changes in the health care system have created difficult and even crisis situations in the lives of HCWs. Crisis situations as states of disorganization cause a person to experience a sense of fear, shock, emotional and psychological destabilization, and difficulties in getting through specific situations [11]. During the coronavirus pandemic, HCWs have faced crisis situations that increased the risk of physical and psychological suffering, conducive to the development of symptoms of anxiety, depression, and other emotional crises [12], as well as psychological disorders manifesting in states of anxiety, panic, or emotional disorders [13]. A review of the mental health literature related to the COVID-19 pandemic reveals preliminary evidence suggesting that symptoms of anxiety and depression and reports of stress are common psychological reactions to the COVID-19 pandemic [14]. In a systematic umbrella review of the global evidence of seven meta-analyses on the incidence of anxiety and depression among HCWs during the COVID-19 pandemic, it was shown that the overall incidence of states of anxiety or depression in HCWs during the COVID-19 epidemic were 24.94% and 24.83%, respectively [15]. In a systematic umbrella review of global evidence comprising ten systematic reviews, on the other hand, it was found that in the group of HCWs, the incidence of anxiety among nurses ranged from 22.8% to 27% while the incidence of depression among nurses was 28% [16]. 

The COVID-19 pandemic has put HCWs around the world in an unprecedented situation, but the risk of adverse psychological effects is particularly high among nurses. Healthcare workers, including large numbers of nurses, face difficult conditions and limited resources in caring for COVID-19 patients, putting them at an increased risk of depression and anxiety [14]. Therefore, it is essential to continue assessing the mental health of nurses and other high-risk groups at the forefront of this pandemic. The scale of this phenomenon changes over time, so it becomes increasingly important to understand the extent of nurses’ mental health problems and needs, and to recognize the nature of these changes, in order to provide mental health services and implement effective psychological interventions [17].

The fast-paced and changeable nature of the mental health emergency during the COVID-19 pandemic and the numerous studies from various countries on the most common mental health problems among nurses that have been published in recent months are prerequisites for systematic reviews. For that reason, the goal of the present study is to update and refine the results of current systematic reviews and meta-analyses published by Olaya et al. [18], Al Maqbali et al. [19], Fernandez et al. [16], and Varghese et al. [20], and to carry out a systematic review and meta-analysis of published studies in the long term over the first eleven months of the COVID-19 pandemic that pertain to the prevalence of anxiety and depression among nurses only. Therefore, this systematic review and meta-analysis had the goal of determining any spread of mental health problems in terms of incidence of depression and anxiety among nurses during the COVID-19 pandemic, with respect to standard measurement tools and taking into account the results of long-term (eleven months) studies assessing the severity of mental health disorders.

We focused our review and meta-analysis on nurses only, the long duration of the pandemic (i.e., eleven months), the use of standardized measurement tools for anxiety and depression, and factors that could be selected relating to the incidence of depression and anxiety, such as gender, marital status, test sites (Asia vs. other continents), and places of employment during the COVID-19 pandemic (first line vs. mixed) of the respondents.

## 2. Materials and Methods

### 2.1. Study Design

This study was conducted in accordance with the recommendations of Preferred Reporting Items for Systematic Reviews and Meta-analyses (PRISMA) [21]. The systematic review protocol is not available in any databases and is available from the authors. The PRISMA checklist for this study is available in Appendix A.

### 2.2. Search Strategy

Relevant articles from the moment the WHO announced the coronavirus pandemic, i.e., from 11 March 2020 to 18 February 2021, were searched for in the PubMed, Web of Science, and SCOPUS databases. Only English-language articles were sought. The article search was performed using an alternate combination of, and/or, the following terms: “COVID”, “COVID-19”, “Severe Acute Respiratory Syndrome Coronavirus 2”, “SARS-CoV-2”, “SARSCoV2”, “SARS CoV2”, “2019-nCov”, “2019 Novel Corona virus”, “Coronavirus Disease 2019”, “Coronavirus Disease-19”, “SARS Coronavirus 2”, “nurses”, “nursing personnel”, “registered nurses”, “nursing staff”, “mental health”, “mental health disorders”, “stress”, “stress disorder”, “post-traumatic stress”, “PSTD”, “mental wellbeing”, “psychological distress”, “depression”, and “anxiety”. The search strategy used for each database is given in the Appendix A. After key articles were identified, manual document searches and tracking were performed for each reference on the list of key articles to increase the sensitivity of the literature search. In the case of a more comprehensive search, there would have been no limit to the results.

### 2.3. Selection Criteria

The selection criteria for relevant articles were the following: (1) the articles were cross-sectional, cohort, or case-control studies; (2) the participants were nurses working in various healthcare facilities during the coronavirus pandemic; (3) the articles were written in English; (4) the articles were published in peer-reviewed journals; (5) the degree of depression or anxiety was measured with a standardized questionnaire; and (6) the studies had sufficient data for calculating the degree of depression or anxiety in groups of practicing nurses. 

Studies were excluded if (1) they were reviews, commentaries, editorials, or summaries of conferences; (2) they were not concerned with the goals of our review; (3) they were conducted on a small group of fewer than twenty respondents; (4) they included representatives of various medical professions without the possibility of extracting separate results for nurses; or (5) there were no clear cut-off points for standardized tools measuring the severity of depression and anxiety, and mean results were not taken into account. 

Two independent reviewers reviewed the titles and abstracts and then the full text of potential articles, in accordance with the inclusion and exclusion criteria. Any discrepancies were established by consensus with a third reviewer.

### 2.4. Data Extraction

The data were extracted by one independent reviewer using predefined data extraction forms. The extracted data were then verified by a second reviewer. All disagreements were resolved by consensus with a third reviewer. The information extracted included characteristics of the study (author, year, and country of publication, study design, and sample size), characteristics of the samples (gender, age, marital status, place of work during the coronavirus pandemic), estimation of the dissemination of depression and anxiety, diagnostic criteria for depression and anxiety, and the research tool used to assess depression and anxiety.

### 2.5. Quality Assessment

Two independent authors assessed the risk of bias of the included studies. Discrepancies were resolved, by consensus, with a third author. In the analyzed cross-sectional studies, the evaluation form recommended by the Agency for Healthcare Research and Quality [22] was used. The checklist consisted of eleven items. Each item was rated as “yes”, “no”, or “unclear”. One point was assigned for each item, if the research met the methodological standards. For ratings of “no” or “unclear”, zero points were assigned. Results with a score of zero to three points indicated a low-quality study, four to seven points indicated a moderate quality study, and eight to eleven points indicated a high-quality study.

### 2.6. Ethical Approval

No ethical approval was obtained for the study because we used published data that had already been ethically validated.

### 2.7. Statistics Analysis

Pooled prevalence of depression and anxiety were calculated using meta-analytic methods. Modeling with random effects and the restricted maximum likelihood (REML) estimator was used to account for between-study heterogeneity. *I*^2^ statistics were calculated to provide a measure of the proportion of overall variation attributable to between-study heterogeneity. Differences in response rate between categories of study definition, number of drugs, and number of types of malignancies were assessed using the Q test for heterogeneity in meta-regression. To examine the influence of several included characteristics on the prevalence of depression and anxiety, we performed a meta-regression. The following factors were studied: percentage of women, percentage of respondents who were married, the place where the study was conducted (Asia vs. other continents), and the place of work during the COVID-19 pandemic (front lines vs. mixed). The Egger test was used to assess the possibility of publication bias. Meta-analysis was conducted using meta for package (R version 3.3.3); *p* < 0.05 was considered statistically significant.

## 3. Results

### 3.1. Study Inclusion

The literature search process and the process for selecting studies are detailed in Figure 1. A total of 3367 studies were identified in the databases. After removing duplicates, 2569 were searched, 2417 of which were then excluded by study title and 112 of which were excluded on the basis of the abstract, leaving 40 studies for the eligibility phase. The full text of these articles was assessed by independent reviewers for eligibility. Fifteen articles were excluded at this stage because they did not meet eligibility requirements for the reasons listed in Figure 1. Following a critical evaluation of the full texts of the articles by consensus, two articles were excluded because it was not possible to define the incidence of depression and anxiety in the analyzed research. Moreover, the decision was made at this stage to include for further analysis only part of the results in an article that evaluated the level of anxiety and depression in nurses during the period of exacerbation and during the stable period of the COVID-19 pandemic [23]. As that article’s research for both periods of the pandemic was conducted with the same group of nurses, it was decided to include only the results obtained during the exacerbation period of the pandemic. In the end, twenty-three studies were used in the qualitative analysis, eighteen of which analyzed the level of depression and twenty-two of which analyzed the level of anxiety.

### 3.2. Description of Included Studies

A summary of the studies selected for our meta-analysis is presented in Table 1. The total sample size analyzed in the twenty-three studies that were included was 44,165 nurses, with sample sizes of each study ranging from 88 to 21,119 participants. With respect to the gender of the participants, one study did not give information on the gender of its participants [24]; this study included 3676 nurses. Of the 40,489 nurses in studies where the percentage of men and women was specified, 95.81% (*n* = 38,792) were women. In fifteen of the studies [6,25,26,27,28,29,30,31,32,33,34,35,36,37,38], the average age of respondents was determined by the mean, which ranged from 28.8 to 45.1 years; in the remaining studies [9,23,39,40,41,42,43], the age distribution was described by categories. In one study [24], neither average age nor age categories was given. 

The research was conducted in nine countries: China (56.52%, *n* = 13) [23,26,27,28,29,32,34,35,38,39,40,41,42], the Philippines (8.7%, *n* = 2) [30,31], the United States (8.7%, *n* = 2) [6,9], Turkey (4.35%, *n* = 1) [36], Saudi Arabia (4.35%, *n* = 1) [25], Iran (4.35%, *n* = 1) [33], Great Britain (4.35%, *n* = 1) [37], Brazil (4.35%, *n* = 1) [43], and Canada (4.35%, *n* = 1) [24]. In eleven of the studies, nurses working on the front lines were surveyed [26,28,30,31,32,34,36,39,40,41,43], while in the other studies the nurses were working on the front lines as well as behind the lines in the fight against COVID-19, without distinct categories.

### 3.3. Quality Assessment

The studies were scored with the aid of the AHRQ checklist. In evaluating the quality of the analyzed studies according to AHRQ assessment criteria, as many as fifteen of the studies were characterized as being of moderate quality, while the remaining eight were characterized as being of low quality. None of the studies received scores indicative of high quality. Detailed results of the evaluation of the quality of the studies included in this meta-analysis are presented in the Appendix A.

### 3.4. Characteristics of Instruments Used to Assess Depression and Anxiety Levels

The characteristics of the nine tools used to assess the severity of depression and anxiety symptoms and the cut-off points for the scales adopted in this meta-analysis are presented in Table 2.

In thirteen of the studies [6,9,23,24,25,26,32,33,34,35,37,39,41] the Patient Health Questionnaire (PHQ-9) for evaluating the severity of depression symptoms was used. In four of the studies [27,28,29,42], the Self-Rating Depression Scale (SDS) was used to assess depression symptoms.

Six tools for assessing the severity of anxiety symptoms were systematically identified among the studies included in the review. In twelve studies [6,9,23,24,25,32,33,34,35,37,39,41] this tool was the Generalized Anxiety Disorder (GAD-7) assessment. In four studies [27,28,29,42] the Self-Rating Anxiety Scale (SAS) was used. In two of the studies [30,31], the Coronavirus Anxiety Scale (CAS) was used to assess the severity level of anxiety. In one study, [40] the Hamilton rating scale for anxiety (HAMA) was used, and one study [36] used the State-Trait Anxiety Inventory (STAI).

Two articles identified two tools consisting of two subscales that assessed both depression and anxiety symptoms at the same time. In one of these articles [38], this tool was the short version of the Depression Anxiety Stress Scale (DASS-21). In the other article, the Hospital Anxiety and Depression Scale (HAD) was used [43].

### 3.5. Incidence of Depression among Nurses during the COVID-19 Pandemic

Figure 2 presents the percentage for occurrence of depression estimated by the analyzed studies. The incidence of depression among nurses during the COVID-19 pandemic was reported in eighteen of the studies, totaling 39,430 respondents. The combined overall rate of depression as assessed by all tools was 22% (95% CI 0.15–0.30, *I*^2^ = 99.71%). The depression rate in thirteen of the studies (14,347 nurses), assessed with the PHQ-9 tool, was 23% (95% CI 0.15–0.31, *I*^2^ = 99.27%). In the remaining five studies (*n* = 25,083 nurses), using other tools (SDS, HAD-D and DASS-21 Depression), the depression rate was 21% (95% CI 0.03–0.38, *I*^2^ = 99.87%).

### 3.6. Incidence of Anxiety among Nurses during the COVID-19 Pandemic

Figure 3 presents the percentage of the occurrence of anxiety estimated in the analyzed studies. Twenty-two of the studies assessed the intensity of anxiety during the COVID-19 pandemic among 43,062 nurses. The overall percentage of respondents with anxiety disorders was 29% (95% CI = 0.18–0.40, *I*^2^ = 99.92%). In twelve of the studies (*n* = 13,244), the severity of anxiety was assessed on the GAD-7 scale, and the percentage of persons with anxiety was 22% (95% CI = 0.14–0.31, *I*^2^ = 99.42%). In four of the studies, anxiety was assessed on the SAS scale (*n* = 27,930). The percentage of respondents with anxiety assessed by the SAS scale was 7% (95% CI = 0.02–0.12, *I*^2^ = 99.49%). For the other 1888 nurses who were evaluated for anxiety by CAS, HAMA, SAI, HAD-A, and DASS-21, the incidence of anxiety was assessed at 57% (95% CI = 0.35–0.79, *I*^2^ = 99.30%).

### 3.7. Factors Relating to the Incidence of Depression and Anxiety

A percentage of the women participating in this study was significantly and negatively associated with the incidence of depression (b = −0.008, 95% CI = −0.013–0.0036, *p* = 0.01) and anxiety (b = −0.012, 95% CI = −0.020–0.0047, *p* = 0.039) in the research reporting the incidence of both depression and anxiety. There was no significant relationship between the percentage of married respondents in the study and the incidence of depression (b = −0.004, 95% CI = −0.0133–0.0052, *p* = 0.39) or anxiety (b = −0.0049, 95% CI = −0.016–0.006, *p* = 0.38). A similar lack of relationships was observed when the studies were narrowed down, with depression and anxiety assessed only via PHQ-9 or GAD-7.

Table 3 presents a comparison of the percentage of depression and anxiety in relation to the location where the study was conducted (Asia vs. the other continents) and the respondents’ places of work during the COVID-19 pandemic (frontlines vs. mixed). No significant percentage difference was observed in depression or anxiety between those working on the frontlines and those behind the lines. There was no significant correlation between the place where the research was carried out and the frequency of anxiety.

### 3.8. Publication Bias

The funnel plot demonstrated a mild asymmetry in prevalence of depression and anxiety (Appendix A). However, the *p*-value for the Egger’s test was 0.32 and 0.10 for depression and anxiety, respectively, indicating no, or undetected, publication bias.

## 4. Discussion

During the pandemic, the high risk of infection and the spread of COVID-19 increased the physical and mental burden of all healthcare workers, including nurses active in their profession [39]. With respect to psychological disorders, nurses are seen as a high-risk group even when working without the additional burden of a pandemic [44,45]. A state of poor mental health in nurses can be detrimental not only to nurses themselves but can also affect the quality of patient care [45]. Unfortunately, nursing shortages are an ongoing problem the world over. High turnover and absenteeism due to illness can cause staff overload and an inability to meet the expectations of patients and their families. An over-taxed team can put patients at risk of a greater error rate, longer hospitalization, and even mortality [46]. Thus, as the prevalence of mental health disorders among nurses increases, so will the economic, social, and individual effects of these disorders [45]. The SARS-CoV-2 pandemic has put a heavy burden on healthcare systems all around the world. To broaden our understanding of the experiences of nurses during the COVID-19 pandemic, we conducted a systematic literature review and meta-analysis of the prevalence of mental health problems in the areas of depression and anxiety disorders in the nursing population during the COVID-19 pandemic, based on the long period of time (eleven months) for which data on the topic has been collected.

The pooled analysis of the data consisted of cross-sectional studies in an overall group of 44,165 nurses. The goal of our systematic review and meta-analysis was to determine the prevalence of anxiety- and depression-related mental health problems among nurses during the COVID-19 pandemic. Severity levels of anxiety and depression were assessed in twenty-three studies [6,9,23,24,25,26,27,28,29,30,31,32,33,34,35,36,37,38,39,40,41,42,43], the results of which were analyzed. Four articles addressed the incidence of anxiety itself, as well as the factors intensifying it, among working nurses [30,31,36,40]. Meta-analysis indicates that the incidence of depression among nurses during the COVID-19 pandemic is 22% (95% CI = 0.15–0.30, *I*^2^ = 99.71%), and the incidence of anxiety is 29% (95% CI = 0.18–0.40, *I*^2^ = 99.92%). The considerable heterogeneity of these results should be emphasized. Our discoveries highlight an important issue regarding nurses and their mental health issues during the COVID-19 pandemic.

An extensive analysis of the literature showed that several systematic reviews and meta-analyses on the incidence of mental health disorders, among HCWs and in the general population, have already been published. Saragih et al. [47] conducted a systematic review and meta-analysis of studies on the spread of anxiety and depression among healthcare workers. In the studies they analyzed, 27.9% of the participants were doctors, 43.7% nurses, and 7% were other kinds of healthcare workers. They determined that the incidence of anxiety among healthcare workers was 39% and the incidence of depression was 36%. A similar systematic review and meta-analysis was carried out by Hao et al. [48]. Their results indicated an incidence of depression and anxiety in healthcare workers of 24.1% and 28.6%, respectively. 

In contrast, the results of a large-scale meta-analysis of evidence summarizing seventy-one published articles on mental health problems during the COVID-19 pandemic, covering a group of 146,139 people from China, the United States, Japan, India, and Turkey, and including patients with confirmed COVID-19 infections, healthcare workers, and the general population, showed a frequency of anxiety symptoms at 32.6% and a frequency of depressive disorders at 27.6% during the COVID-19 pandemic. The authors of that meta-analysis also made the noteworthy observation that mental health problems (i.e., anxiety and depression) had the highest incidence in COVID-19 patients, and that lower levels of anxiety and depression, as well as sleep problems, were observed in healthcare professionals than in the general population. Another systematic review and meta-analysis found that the prevalence of depression among all healthcare workers was 24% (95% CI = 20–28%), while among nurses it was 25% (95% CI = 18–33%), among doctors it was 24% (95% CI = 16–31%), and for frontline healthcare workers it was 43% (95% CI = 28–59%) [18].

Al Maqbali et al. [19] conducted a systematic review and meta-analysis of the spread of stress, depression, anxiety, and sleep disorders among nurses during the COVID-19 pandemic. Their results indicated that the incidence of anxiety was 37% (95% CI = 32–41%), while the incidence of depression was 35% (95% CI = 31–39%). In another systematic review and meta-analysis that assessed the spread of mental health disorders among nurses, it was found that the incidence of anxiety symptoms among the surveyed nurses was 33% (95% CI = 24–43%) with significant heterogeneity (I^2^ = 99.4%, *p* < 0.01), while the occurrence of depression was 32% (95% CI = 21–44%), with significant heterogeneity (I^2^ = 99.4%, *p* < 0.01) [20]. 

In our meta-analysis, we found a lower incidence of anxiety and depression symptoms than in the studies cited above. The higher incidence of anxiety and depression in the meta-analyses conducted by Saragih et al. [47] and by Liu et al. [49], in comparison with our own results, may be related to distinctions in the work carried out in the various professions among the respondents. The duration of data collection in these studies also has significance. Higher rates of anxiety and depression, in comparison with our results, were also demonstrated in meta-analyses carried out on groups of nurses, i.e., in the meta-analyses of Al. Maqbali et al. [19] and Varghese et al. [20]. Our systematic review and meta-analysis summarize publications from both the onset of the pandemic and the period that followed (covering eleven months), compared to the cited systematic reviews and meta-analyses. It is therefore worthwhile to compare the results obtained in our meta-analysis with the results of meta-analyses published in this field before the COVID-19 pandemic. Unfortunately, there are not many such publications in the literature. Results of a systematic review and meta-analysis of the incidence of depression among nurses of various departments and hospitals in Iran indicated the prevalence of depression symptoms as being at a level of 26.88% (95% CI = 21.45–31.91%) [50]. A trend of change in the rate of incidence of depression can therefore be observed in this example, taking into account the time of publication of data from the various meta-analyses within the narrative analysis. However, a hypothesis that chronic effect factors related to the COVID-19 epidemic are a self-regulating restraint on the incidence of depressive and anxiety disorders among nurses cannot constitute a prerequisite for outreach and intervention studies for alleviating mental health problems among nurses and helping them cope with their burdens.

The prolonged duration of the COVID-19 pandemic also prompts a search for evidence of its long-term psychological effect on HCW. A replication cross-sectional study one year after the COVID-19 outbreak to assess the mental health outcomes of HCW (*n* = 1 033) at an academic hospital in Verona (Italy) found that the percentage of HCW above the cut-off point increased from 2020 to 2021 across all performance domains (anxiety, 50.1% vs. 55.7, *p* < 0.05; depression, 26.6% vs. 40.6%, *p* < 0.001). In turn, a multivariate analysis showed that one year after the COVID-19 outbreak, nurses were more likely to experience anxiety and depression than other HCWs [51].

The measurement tools used to assess the prevalence of anxiety and depression, namely their psychometric properties and the cut-off points that were adopted, may be important factors in the differences identified in meta-analyses. In our research, we sought to select studies on the basis of depression and anxiety being measured by the use of standardized questionnaires, and we made critical quality assessments of the studies that were included. In terms of moderation, analysis of a large-scale meta-analysis of the evidence [49] of the moderating role of measuring tools on the results in assessing mental health problems among research participants during a pandemic was confirmed, with results varying significantly depending on the scale used. Thus, the high prevalence of mental health problems during the COVID-19 pandemic, including anxiety symptoms and depressive disorders, may indicate (when analyzed in a fairly large sample) that heterogeneous results for these mental health problems may be caused by the use of non-standardized tools without reliable psychometric properties being maintained for the studied populations. Another possible reason for the differences in prevalence is the variation in the cut-off points for elevated symptoms for the same measurement instrument of the studied variable, which we took into account when qualifying studies for our analysis (for details, see Table 1).

In addition, research results published in scientific journals have provided very timely and significant evidence that the COVID-19 pandemic poses a threat to the mental health of individuals. However, it should be noted that most of these studies were conducted in the early and peak periods of the pandemic’s development, which may indicate an overestimation of the frequency of these problems. Furthermore, in the interests of sharing new research results in a timely manner, articles that were not of high quality have been published in some journals. Our quality assessment using the AHRQ checklist showed that eight of the twenty-three articles included in the review were of low quality. It should be noted that all studies included in our meta-analysis used self-reported standardized questionnaires to assess symptomatology of depression and anxiety. Moreover, the use of a large variety of scales could lead to differences in the assessment of depression and anxiety occurrence. In fact, our results showed that studies using the SDS (Self-Rating Depression Scale) questionnaire indicated lower rates of depression, while studies using CAS, HAMA, SAI, HAD-A, and DASS-21 indicated a much higher prevalence of anxiety. Despite the convenience of using the same standardized measurement tools for an initial assessment of the characteristics of a diagnosis based on clinical interviews, it is not always possible for this usage to be fully reflected in epidemiological studies, because these are simply screening tools and require in-depth clinical diagnostic follow-up.

### Study Limitations

When interpreting our results, it is necessary to take certain limitations into account. First, most of the studies included in the meta-analysis used convenient samples, so their representativeness of the nursing population may be unreliable. Second, depression and anxiety were evaluated mainly by using self-reported data from questionnaires that might also introduce other psychological and emotional manifestations, e.g., strongly expressing public approval of the medical profession during the pandemic. Such data may also be less accurate than data from full clinical interviews. Third, an assessment of the incidence of depression and anxiety among nurses in professional practice, based on their inclusion in cross-sectional studies in the meta-analysis, makes it difficult to establish a causal relationship between the pandemic and depression and anxiety. Depressive disorders and anxiety symptoms in the included studies had not been assessed before the pandemic. This has limited our ability to investigate additional psychological strains on nurses caused by the COVID-19 pandemic, as we do not have data on their prior mental state. For this reason, it is necessary to be careful when interpreting our results. In addition, our systematic review was conducted mainly in medical databases (PubMed, SCOPUS, and Web of Science); therefore, some articles, especially those related to psychology, may not be reflected. Further, our research did not take into account reports that concerned all health care professionals, among which a group of nurses was included as a subset.

Finally, we found certain sources of heterogeneity. In the articles we analyzed, various scales were used and various cut-off values were adopted. For example, the use of the SDS questionnaire was associated with lower rates of depression, and the use of the CAS, HAMA, SAI, HAD-A, and DASS-21 scales were associated with higher rates of anxiety. In addition, the studies we analyzed were carried out at various points of time; the feelings of nurses might differ as the COVID-19 pandemic runs its course from the moment when it first appeared. Many new scientific papers on COVID-19 are published each day, and as the pandemic continues, we have ever-increasing knowledge about it, which makes it difficult to conduct an up-to-date and in-depth review. 

Another limitation is the fact that the vast majority of the studies analyzed in the present review came from Asia, mainly China, with only a small portion coming from other continents. This geographical and cultural context may have influenced various types of psychological responses to the same stressor among healthcare workers; therefore, the obtained results should not be generalized for all nurses. Yet another important limitation is the fact that due to the use of the selected inclusion and exclusion criteria, it is possible that relevant articles may have been omitted in the first stage of collecting the data for this review.

Future studies should strive to investigate the prevalence of depression and anxiety among nurses in other countries and, where possible, use random sampling as well as longitudinal designs for determining the evolution of mental health problems in this population. In addition, subsequent systematic reviews and meta-analyzes could consider the severity of depression and anxiety in nurses at different periods of the pandemic, taking into account milestones in the fight against the pandemic.

## 5. Conclusions

To summarize, our systematic review and meta-analysis provide a long-term and comprehensive synthesis of existing evidence confirming the incidence of depressive disorders in more than one-fifth of those studied, and anxiety symptoms among just under one-third of nurses, during the COVID-19 pandemic. These findings help quantify the emerging need for psychological support for nurses within the context of the COVID-19 pandemic. As new evidence continues to emerge, we will be able to continue updating this meta-analysis and carrying out further efforts in analyzing factors related to the epidemic, to facilitate planning at the national level, improve mental health security system interventions, and design prospective solutions for similar epidemic events involving those in the nursing profession in the future.

## Figures and Tables

**Figure 1 ijerph-19-01154-f001:**
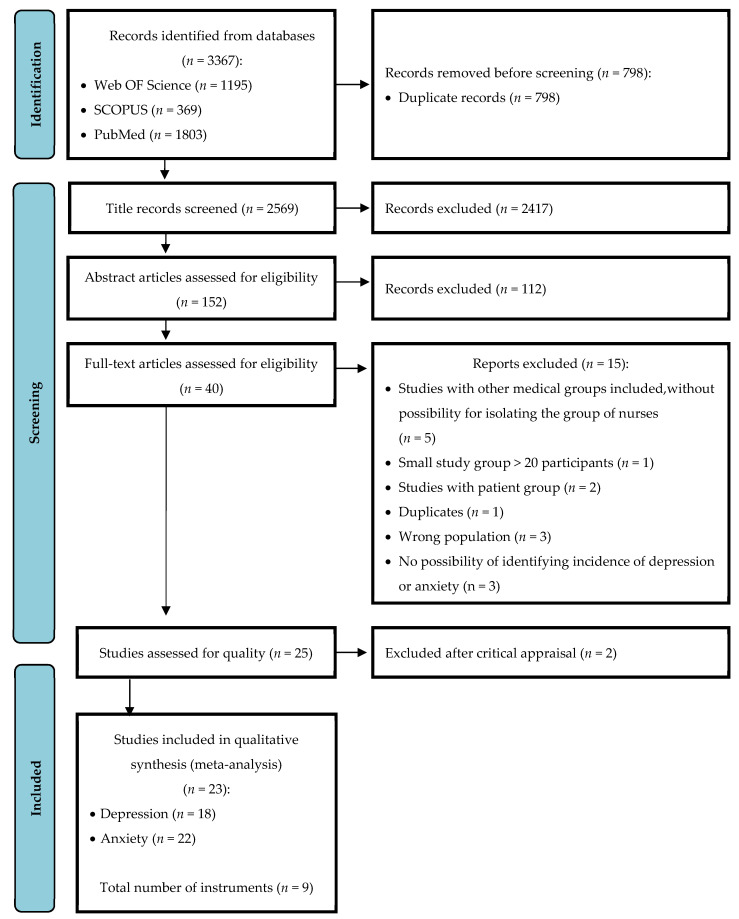
PRISMA flow diagram.

**Figure 2 ijerph-19-01154-f002:**
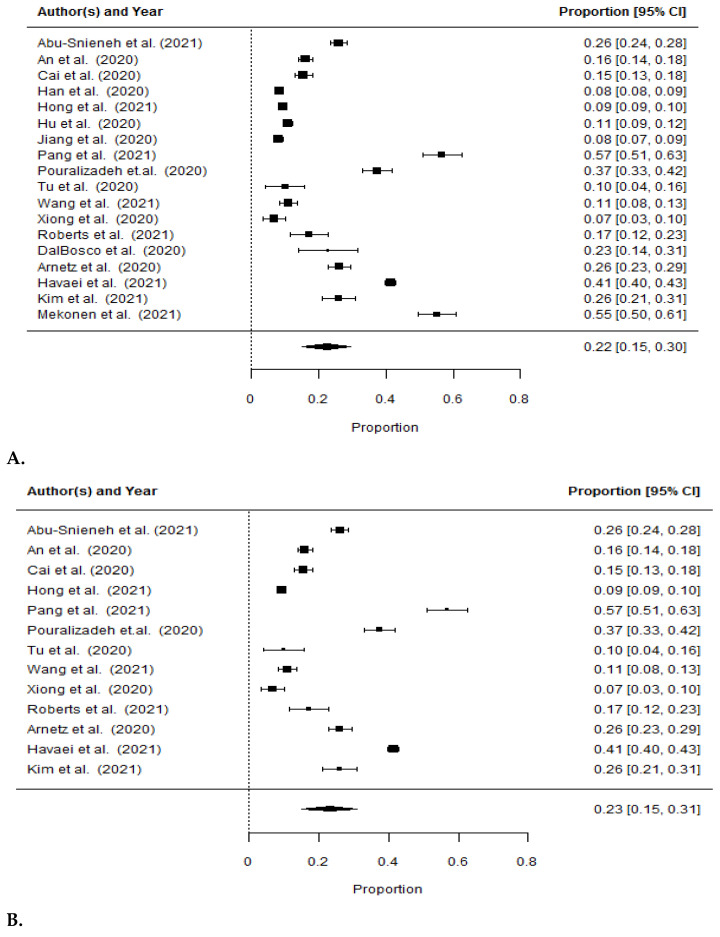
Forest plot for the prevalence of depression among nurses: (**A**) overall result; (**B**) assessment scale PHQ-9; (**C**) remaining assessment scales, excluding PHQ-9.

**Figure 3 ijerph-19-01154-f003:**
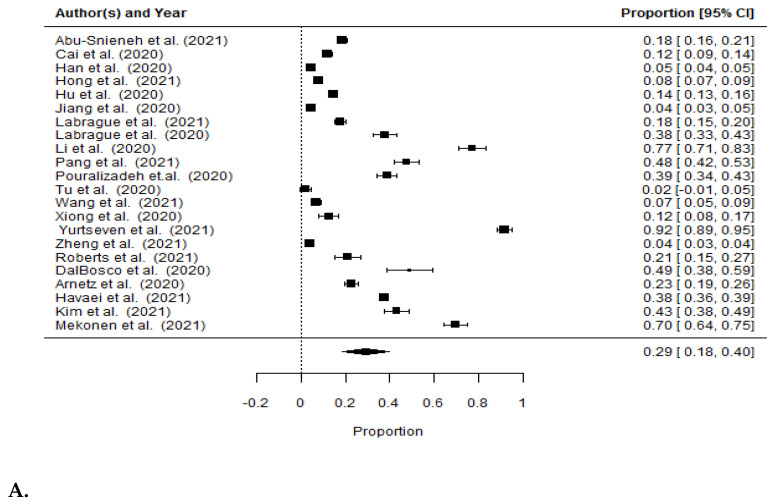
Forest plot for the prevalence of anxiety among nurses: (**A**) overall result; (**B**) assessment scale GAD-7; (**C**) assessment scale SAS; (**D**) remaining assessment scales, excluding GAD-7 and SAS.

**Table 1 ijerph-19-01154-t001:** Characteristics of the included studies.

No.	First Author (Year)/Country	Study Design	Study Size	Participants	Age, Years (Mean ± SD or *n* (%))	Female *n* (%)	Married *n* (%)	Position	Start Date	End Date	Depression Assessment Tool	*n* (%) with Depression	Anxiety Assessment Tool	*n* (%) with Anxiety	Survey Method	Quality Score
1.	Abu-Snieneh (2021)/Saudi Arabia [25]	cross-sectional study	1265	nurses from all regions of Saudi Arabia	28.83 ± 5.29	1101 (87)	783 (61.9)	Mixed	April 2020	June 2020	PHQ-9	329 (25.9)	GAD-7	234 (18.5)	Online survey	5
2.	An et al. (2020)/China [26]	cross-sectional study	1103	Emergency Department nurses from all regions of China	32.20 ± 7.61	1001 (90.8)	710 (64.4)	Frontline	15 March 2020	20 March 2020	PHQ-9	176 (16)	N/A	N/A	Online survey	4
3.	Cai et al. (2020)/China [23]	longitudinal study	709	nurses from Renmin Hospital of Wuhan University (outbreak period of the pandemic)	≥ 30 287 (40.4)	684 (96.5)	376 (53)	Mixed	29 January 2020	2 February 2020	PHQ-9	109 (15.4)	GAD-7	84 (11.8)	Online survey	7
4.	Han et al. (2020)/China [27]	cross-sectional survey	21,119	nurses from 14 prefecture and city hospitals in Gansu Province, located in northwest China	31.89 ± 7.08	20909 (98.6)	15499 (73.1)	Mixed	7 February 2020	10 February 2020	SDS	1738 (8.2)	SAS	996 (4.7)	Online survey	5
5.	Hong et al. (2021)/China [39]	cross-sectional study	4692	nurses from the Chongqing region	≥ 31 2043 (43.6)	4548 (96.9)	3013 (64.2)	Frontline	8 February 2020	14 February 2020	PHQ-9	442 (9.4)	GAD-7	379 (8.1)	Online survey	4
6.	Hu et al. (2020)/China [28]	cross-sectional study	2014	nurses from two hospitals in Wuhan	30.99 ± 6.17	1754 (87.1)	1230 (61.1)	Frontline	13 February 2020	24 February 2020	SDS	217 (10.7)	SAS	288 (14.3)	Online survey	5
7.	Jiang et al. (2020)/China [29]	cross-sectional study	1569	nurses from the Linxia Hui Autonomous Prefecture	30.93 ± 6.48	1550 (98.8)	1170 (74.6)	Mixed	6 February, 2020	10 February 2020	SDS	127 (8.1)	SAS	68 (4.3)	Online survey	3
8.	Labrague et al. (2021)/Philippines [31]	cross-sectional study	736	nurses from frontline hospital and public health nurses in Western Samar	31.9 ± 7.35	574 (78.1)	312 (42.45)	Frontline	1 September 2020	1 October 2020	N/A	N/A	CAS	130 (37.4)	Online survey	3
9.	Labrague et al. (2020)/Philippines [30]	cross-sectional study	325	nurses from frontline hospital in Region 8, Philippines	30.94 ± 6.67	243 (74.8)	108 (33.2)	Frontline	25 April 2020	25 May 2020	N/A	N/A	CAS	123 (37.8)	PAPI	4
10.	Li et al. (2020)/China [12]	cross-sectional study	176	nurses from tertiary hospitals in Wuhan city, Hubei province that were designated to receive new patients with COVID-19	≥30 72 (40.8)	136 (77.3)	88 (50)	Frontline	Unknown	Unknown	N/A	N/A	HAMA	136 (77.3)	Online survey	3
11.	Pang et al. (2021)/China [32]	cross-sectional study	282	nurses from three hospitals that received patients with COVID-19 in both Guangdong and Hubei Provinces	31.61 ± 7.60	250 (88.65)	169 (59.93)	Frontline	10 March 2020	20 March 2020	PHQ-9	160 (56.74)	GAD-7	134 (47.52)	Online survey	4
12.	Pouralizadeh et al. (2020)/Iran [33]	cross-sectional study	441	Nurses working in the province of Guilan at the University of Medical Sciences hospital	36.34 ± 8.74	420 (95.2)	335 (76)	Mixed	7 April 2020	12 April 2020	PHQ-9	165 (37.5)	GAD-7	171 (38.7)	Online survey	3
13.	Tu et al. (2020)/China [34]	cross-sectional study	100	nurses from Wuhan in “Huoshenshan” hospital	34.44 ± 5.85	100 (100)	70 (70)	Frontline	7 February 2020	24 February 2020	PHQ-9	10 (10)	GAD-7	2 (2)	Online survey	4
14.	Wang et al. (2021)/China [35]	cross-sectional study	586	nurses working in Nanjing in the province of Jiangsu	31.07 ± 7.54	563 (96.08)	353 (60.24)	Unknown	14 February 2020	3 March 2020	PHQ-9	64 (11.09)	GAD-7	40 (6.83)	Online survey	6
15.	Xiong et al. (2020)/China [41]	cross-sectional study	223	nurses from one of the public tertiary hospitals in Xiamen, Fujian Province	≥36 77 (34.5)	217 (97.3)	Unknown	Frontline	16 February 2020	25 February 2020	PHQ-9	15 (6.7)	GAD-7	27 (12.1)	Online survey	5
16.	Yurtseven et al. (2021)/Turkey [36]	cross-sectional study	270	Nurses working in a university hospital operating as a pandemic hospital	36.83 ± 9.23	237 (87.77)	193 (71.48)	Frontline	Unknown	Unknown	N/A	N/A	SAI	249 (92.4)	Online survey	3
17.	Zheng et al. (2021)/China [42]	cross-sectional study	3 228	nurses from Sichuan Province and Wuhan City	≥30 1706 (52.9)	3121 (96.7)	Unknown	Mixed	27 January 2020	3 February 2020	SDS other cutoff points	N/A	SAS	122 (3.8)	Online survey	4
18.	Roberts et al. (2021)/United Kingdom [37]	cross-sectional study	255	nurses working in respiratory clinical areas	45.1 ± 9.77	226 (88.6)	Unknown	Mixed	1 May 2020	1 June 2020	PHQ-9	31 (17.2)	GAD-7	40 (20.9)	Online survey	3
19.	Dal’Bosco et al. (2020)/Brazil [43]	cross-sectional study	88	nurses working at a regional university hospital of reference for coping with COVID-19 in Paraná	≥ 31 51 (58)	79 (89.8)	32 (36.4)	Frontline	March 2020	April 2020	HAD-D	22 (25)	HAD-A	43 (48.9)	Online survey	5
20.	Arnetz et al. (2020)/USA [9]	cross-sectional study	695	nurses working in the state of Michigan	≥45 376 (54.7)	644 (93.6)	Unknown	Mixed	7 May 2020	29 May 2020	PHQ-9	167 (26.1)	GAD-7	144 (22.6)	Online survey	3
21.	Havaei et al. (2021)/Canada [24]	cross-sectional study	3 676	members of the provincial nurses’ union	Unknown	Unknown	Unknown	Mixed	January 2020	July 2020	PHQ-9	1391 (41.4)	GAD-7	1273 (37.6)	Online survey	3
22.	Kim et al. (2021)/USA [6]	cross-sectional study	320	nurses who graduated from the nursing school at a private, 4-year liberal arts university in Southern California	33 (min-max: 21–67) BRAK SD	302 (94.4)	Unknown	Mixed	20 April 2020	10 May 2020	PHQ-9	83 (26)	GAD-7	138 (43)	Online survey	5
23.	Mekonen et al. (2021)/China [38]	cross-sectional study	293	nurses working in the northwest of Amhara Regional	29.6 ± 5.1	133 (45.4)	156 (53.2)	Mixed	25 September 2020	20 October 2020	DASS-21 Depression	162 (55.3)	DASS-21 Anxiety	204 (69.6)	PAPI	6

Abbreviations: PHQ-9: 9-item Patient Health Questionnaire; GAD-7: 7-item Generalized Anxiety Disorder; SDS: Self-Rating Depression Scale; SAS: Self-Rating Anxiety Scale; CAS: Coronavirus Anxiety Scale; PAPI: Paper and Pencil Interview; HAMA: the Hamilton rating scale for anxiety; BDI: Beck Depression Inventory; SAI: State Anxiety Inventory; DASS-21: Depression Anxiety Stress Scales (short version); HAD: Hospital Anxiety and Depression Scale; HAD-D: Hospital Anxiety and Depression Scale, subscales for identifying depression; HAD-A: Hospital Anxiety and Depression Scale, subscales for identifying anxiety;

**Table 2 ijerph-19-01154-t002:** Characteristics of research tools used in assessing depression and anxiety levels.

Instrument	Name Abbreviation	Characteristic Being Assessed	Administration Method	Number of Items, (Sub)Scale(s) (Number of Items)	Response Options, Range of Score	The Cut-Off Point Adopted in the Meta-Analysis
Patient Health Questionnaire	PHQ-9	Depression	Self-reported	9 items	Dichotomous scoring system and 4-points Likert scale 0–27 - normal (0–4), - mild depression (5–9), - moderate depression (10–14), - severe depression (15–27)	≥10 points
Self-Rating Depression Scale	SDS	Depression	Self-reported	20 items	Dichotomous scoring system and 4-points Likert scale 25–100 - normal (25–52), - mild depression (53–62), - moderate depression (63–72), - severe depression (≥73)	≥63 points
Depression Anxiety Stress Scales	DASS-21	Depression/Anxiety	Self-reported	21 items, on 7 items for each subscale: - depression - anxiety - stress	Dichotomous scoring system and 4-points Likert scale 0–21 for each subscale	Depression ≥ 10 points Anxiety ≥ 8 points
Hospital Anxiety and Depression Scale	HAD	Depression/Anxiety	Self-reported	14 items, on 7 items for each subscale: - HAD-A-anxiety - HAD-D-depression	Dichotomous scoring system 0–21 for each subscale	≥8 points for each subscale:
Generalized Anxiety Disorder	GAD-7	Anxiety	Self-reported	7 items	Dichotomous scoring system and 4-points Likert scale 0–21 - normal (0–4), - mild anxiety (5–9), - moderate anxiety (10–14), - severe anxiety (15–27)	≥10 points
Self-Rating Anxiety Scale	SAS	Anxiety	Self-reported	20 items	Dichotomous scoring system and 4-points Likert scale 25–100 - normal (25–49), - mild anxiety (50–59), - moderate anxiety (60–69), - severe anxiety (70–100)	≥60 points
Coronavirus Anxiety Scale	CAS	Anxiety	Self-reported	5 items	Dichotomous scoring system and 5-points Likert scale 5–25	≥9 points
The Hamilton rating scale for anxiety	HAMA	Anxiety	Self-reported	14 items	Dichotomous scoring system 0–56	≥7 points
State Anxiety Inventory	SAI	Anxiety	Self-reported	20 items	Dichotomous scoring system and 4-points Likert scale 20–80 - normal (≤36), - moderate anxiety (37–42), - high anxiety (≥43)	≥37 points

**Table 3 ijerph-19-01154-t003:** Factors related to the incidence of depression and anxiety.

Variable	Depression			Anxiety		
	Proportion	95% CI	*p*	Proportion	95%CI	*p*
Place of research:						
	All instruments					
Asia	0.18	(0.09–0.26)	0.063	0.25	(0.11–0.38)	0.196
Other continents	0.31	(0.20–0.43)		0.40	(0.20–0.61)	
	Only PHQ-9			Only GAD-7		
Asia	0.21	(0.10–0.31)	0.44	0.18	(0.07–0.29)	0.15
Other continents	0.28	(0.13–0.42)		0.31	(0.17–0.45)	
Position:						
	All instruments					
Frontline	0.19	(0.07–0.31)	0.45	0.31	(0.15–0.46)	0.43
Mixed	0.25	(0.15–0.35)		0.23	(0.11–0.35)	
	Only PHQ-9			Only GAD-7		
Frontline	0.20	(0.06–0.33)	0.54	0.17	(0.02–0.32)	0.40
Mixed	0.25	(0.18–0.33)		0.26	(0.16–0.35)	

## Data Availability

The datasets used and/or analyzed during the current study are available from the corresponding author on reasonable request.

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
