# Peer review of "Prevalence of Depression and Anxiety in Nurses during the First Eleven Months of the COVID-19 Pandemic: A Systematic Review and Meta-Analysis"

_ijerph, 2022, doi:10.3390/ijerph19031154_

Round 1

Reviewer 1 Report

What is the justification for choosing the topic of the systematic review and meta-analysis undertaken by the Authors:

  • Nurses, due to the nature of their work, are particularly exposed to a high risk of mental health disorders during the COVID-19 pandemic
  • In addition to growing staff shortages of nurses, mental health has become another major concern in the group of nurses during the global COVID-19 pandemic.
  • Nurse`s work in a prolonged pandemic conditions will cause both physical and mental exhaustion

What new the article brings to the current knowledge:

  • Meta-analysis shows the prevalence of depression and anxiety among nurses over the long 11-month period of the assessment of mental health problems.
  • More than one fifth of the nurses working during the COVID-19 epidemic experienced depressive disorders and almost one third experienced symptoms of anxiety.
  • Specific intervention studies are needed to alleviate mental health problems among nurses and help them cope with their burden

Before publishing the article, minor additions, editorial corrections and correction of stylistic errors should be made:

  1. In the date notation, I propose to insert commas, for example, in the abstract: lines 15-16 is March 3 2020, I propose to change it to March 3, 2020. Similarly, lines 102-103.
  2. In Table 1, I propose to remove the cut-off points adopted for the tools, because this information is found in Table 2, in which the authors describe the measurement tools used in the articles to assess the exacerbation of depression and anxiety. However, in Table 1, I propose to leave the names of the tools used to assess these variables.
  3. Please, standardize the date notation in the text and Table 1.
  4. Lines 306-307: Part of the sentence is probably missing here? Please, explain.
  5. It is worth taking into account in the discussion other current publications from IJERPH, which may already include a specific thematic area undertaken by the Authors.

Reviewer 2 Report

This is an excellent systematic review/meta-analysis to summarize the impact that the first year of the pandemic had on the nursing workforce in global settings.  The uncovering of widespread anxiety and depression in the nursing workforce is an important first step in identifying the impact that caring for acutely ill patients during a pandemic has had on nurses and what potential interventions will be needed for the future to re-stabilize the nursing workforce.  The authors are transparent about how the studies that were included were selected and the rationale for the exclusion criteria.  The variety of instruments used to measure anxiety and depression are summarized in Table 2 and the authors discuss the limitation that the instruments were "non-standardized" and without "reliable psychometric properties" in their measurement.  The studies that were included in the meta-analysis were outlined in Table 1 with good clarity.  The authors mention the limitation of self-report by participants for the majority of the data collection in those studies.  The write up of the statistical analysis is well done.  The discussion section was extensive and comprehensive and conclusions were appropriate.   In lines 306, 307 there are errors to be removed and the first word (authors) in line 309.  I am very curious that marital status, age, and gender were important characteristics of the sample in the studies but I do not see any discussion in this meta-analysis about whether these variables were found to correlated with the depression/anxiety being experienced in the studies.  Marital status especially seems to be a curious demographic to collect for the studies without an explanation as to it's relevance in the experience of anxiety/depression.  I am curious if there was an increase in anxiety in the younger vs. older population of nurses as well.   I am also curious about a pre-pandemic measurement of anxiety/depression in nursing compared to the pandemic measurements because it may have been high pre-pandemic as well.   Nice work!

Reviewer 3 Report

The manuscript entitled "Incidence of Depression and Anxiety in Nurses During the First 11 Months of the COVID-19 Pandemic: A Systematic Review and Meta-Analysis" covers a very important and timely topic. Overall the manuscript has high quality, therefore I recommend publishing it with minor revisions recommended in the following.

Comments:

  1. The introduction well establishes the argument behind the systematic review of anxiety and depression in medical nurses. On the other hand, the present review could more explicitly name the knowledge gap and the added value of the present meta-analysis, as there were other similar meta-analyses done previously. In particular, the authors state that the goal of the present meta-analysis is to „update and refine the results of current systematic reviews“. However, it is not explicitly stated why the current systematic reviews need refinement. This should be made more explicit.
  1. In the introduction, the authors state that „the scale of anxiety and depression in healthcare nurses changes over time (p. 2)“, which I agree is an important point. Therefore, it would be valuable if authors could address this problem in the data analysis. For example, by comparing the studies from the onset of the pandemic with later studies if it is possible. A comparison like this is currently not included in the study.
  2. In the results (p. 16 & p. 17), the authors mention the effects of other factors, such as gender and marital status, however, it would be more informative if the authors included a standardized effect size as well.
  3. On p. 17, the authors state that "No significant percentage difference was observed in depression or anxiety between those working on the frontlines and those behind the lines. A lower percentage of incidence of depression occurred in Asia, as compared to studies conducted in other countries, but the difference as only borderline statistical significance (p = 0.063)." Here (as well as in attached table 3) the authors state the proportion of anxiety and depression in both groups (with 95%CI), however, they fail to state the difference in proportion and 95% CI of the difference. Merely a p-value for the difference does not provide all the necessary information. Moreover, the concept of "borderline statistical significance" is questionable, therefore I recommend the authors not to use the term.
  4. The first sentence of the discussion does not follow the results the authors presented. Therefore I recommend the authors to rephrase the beginning of the discussion to better follow the analysis above.
  5. The final remark is a minor issue, the authors state in discussion (p.18) "in contrast, results of a large-scale meta-analysis of evidence summarizing seventy-one published articles on mental health problems during the COVID-19 pandemic, covering a population of 146,139 people…" The term population is not accurate here, as this refers to a sample of 146,139 people across several studies, not to a population of this size. The reference population is probably much bigger.
